# Single-Cell Analysis of CHO Cells Reveals Clonal Heterogeneity in Hyperosmolality-Induced Stress Response

**DOI:** 10.3390/cells11111763

**Published:** 2022-05-27

**Authors:** Nadiya Romanova, Julian Schmitz, Marie Strakeljahn, Alexander Grünberger, Janina Bahnemann, Thomas Noll

**Affiliations:** 1Cell Culture Technology, Faculty of Technology, Bielefeld University, 33615 Bielefeld, Germany; marie.strakeljahn@uni-bielefeld.de (M.S.); thomas.noll@uni-bielefeld.de (T.N.); 2Multiscale Bioengineering, Faculty of Technology, Bielefeld University, 33615 Bielefeld, Germany; j.schmitz@uni-bielefeld.de (J.S.); alexander.gruenberger@uni-bielefeld.de (A.G.); 3Institute of Physics, University of Augsburg, 86159 Augsburg, Germany; janina.bahnemann@uni-a.de

**Keywords:** CHO, hyperosmolality, single-cell analysis, mitochondria

## Abstract

Hyperosmolality can occur during industrial fed-batch cultivation processes of Chinese hamster ovary (CHO) cells as highly concentrated feed and base solutions are added to replenish nutrients and regulate pH values. Some effects of hyperosmolality, such as increased cell size and growth inhibition, have been elucidated by previous research, but the impact of hyperosmolality and the specific effects of the added osmotic-active reagents have rarely been disentangled. In this study, CHO cells were exposed to four osmotic conditions between 300 mOsm/kg (physiologic condition) and 530 mOsm/kg (extreme hyperosmolality) caused by the addition of either high-glucose-supplemented industrial feed or mannitol as an osmotic control. We present novel single-cell cultivation data revealing heterogeneity in mass gain and cell division in response to these treatments. Exposure to extreme mannitol-induced hyperosmolality and to high-glucose-oversupplemented feed causes cell cycle termination, mtDNA damage, and mitochondrial membrane depolarization, which hints at the onset of premature stress-induced senescence. Thus, this study shows that both mannitol-induced hyperosmolality (530 mOsm/kg) and glucose overfeeding induce severe negative effects on cell growth and mitochondrial activity; therefore, they need to be considered during process development for commercial production.

## 1. Introduction

For more than 30 years, Chinese hamster ovary (CHO) cells have been the predominant system used by researchers for studying and exploring heterologous protein expression [1]. Their robust growth to high cell densities, easy scalability as suspension-grown cells combined with their genetic pliability, and human-like post-translational modifications are all characteristics that establish these cells as the preferred production platform for numerous pharmaceutical products. To achieve maximal cell density and product titers, CHO production processes are usually run in a fed-batch mode, during which highly concentrated feed and pH-controlling solutions are added. This leads to gradients in pH value and nutrient concentration inside stirred-tank bioreactors [2,3] and results in zones of different osmotic conditions; therefore, there is distinct stress for the cells passing through these zones [4,5]. Hyperosmolality can affect production rates and growth behaviors, but both significantly increased product titer [6,7,8,9] and no observable effects [10,11] have been reported. However, studies consistently report a negative influence of hyperosmolality on cellular growth.

Hyperosmotic exposure negatively impacts the viable cell density (VCD) and causes a pronounced increase in cell size [5,9,11,12,13,14]. These effects are usually explained as being a dose-dependent response to hyperosmotic stimulus, leading to a rather homogenous increase in cell size. However, such mean observations of the population mask potential cellular heterogeneity in response to hyperosmotic exposure. Namely, the observed mean cell size increase could also result from some cells proliferating while others only gain in mass.

Mitochondria are the prime ATP-generating organelles necessary for normal cellular functioning [15,16]. The mitochondrial membrane potential (ΔΨm) is generated via proton pumps which are ordered into complexes I, III, and IV, and which are located between the outer and the inner membranes of the mitochondria which fuel ATP regeneration. Here, the levels of inter-membrane polarization and ATP concentration are kept relatively constant, since their fluctuations may have deleterious effects on the cell. If a cell is exposed to high glucose concentrations, these mitochondrial systems can become overwhelmed and leak protons into the cytoplasm. This leads to the formation of excessive reactive oxygen species (ROS, such as HO^•^ or O^2−^) [17]. Mitochondria play a central role in the cellular stress response and numerous aspects of their involvement in counteracting osmotic changes have been well documented in the relevant literature [8,11,18,19,20,21]. However, consistent observations pinpointing exactly how mitochondrial density and activity change in osmotically stressed CHO cells remain relatively few and far between.

To study the effects of hyperosmolality, a variety of reagents, most prominently NaCl [8,12,22,23], have been used. Sodium chloride is particularly DNA-damaging, limiting its use as a neutral osmotic control [24,25]. In addition, studying osmolality increase via oversupplemented feed, which is more commonly seen in industrial cultivation setups [14,26], may not give researchers a full picture, since various nutrients can have their specific negative effects on cellular growth. Accordingly, any attempts at a comprehensive analysis of the osmotic stress response of CHO cells has been relatively limited to date, since observable effects cannot be related to higher ambient osmolality when active substances, such as NaCl or nutrients, are applied to increase osmotic conditions.

This study elucidated the situation by using a mannitol-supplemented feed to disentangle nutrient-specific and osmotic effects on the cells. Mannitol is one of the most neutral substances and has been used for similar studies using a plethora of other cell types [27]. Mannitol does not seem to be imported into the cells; thus, it remains restricted to the extracellular space [28,29,30].

Recent research of single cells revealed extensive genomic and productivity cell-to-cell heterogeneity even within populations originating from a single clone [31]. This has an extensive relevance to bioprocess performance and stability. Therefore, single-cell data on production-relevant parameters, such as growth and production rate, are in high demand [32]. The heterogeneity of stress responses of suspension-grown CHO cells has not been previously elucidated and there is a high demand for new insights into this aspect to investigate its impact on bioproduction processes.

Our research also targeted the following two primary aspects of this problem: considering cellular heterogeneity in response to hyperosmolality and exploring the role of mitochondria in this process. Therefore, we measured the mitochondrial mass, membrane potential, and mtDNA copy numbers to access the response to a stage-wise increase in osmolality that was created either by adding a highly supplemented feed or mannitol. We also focused on the cellular physiology and the heterogeneity of the cellular stress response by utilizing a microfluidic single-cell cultivation device and comparing the observations with those performed for a bulk population.

## 2. Materials and Methods

### 2.1. Cell Culture Maintenance

For this project, we used a suspension-adapted antibody-producing CHO DP-12 cell line (clone#1934, ATCC CRL-12445), as well as the same CHO DP-12 cell line transduced with a cell cycle-specific fluorescent ubiquitination-based cell cycle indicator (FUCCI) system [33]; the latter was kindly provided by Prof. Dr. Pörtner (Hamburg University of Technology, Hamburg, Germany) [34]. The FUCCI cells were transduced with the cell cycle-specific fluorescent ubiquitination-based indicator system and produced either a green fluorescent protein, mVenus, coupled to a DNA replication inhibitor, hGeminin, expressed when the cell is currently in the S/G2/M phase (measured via an FL1 channel) or a red fluorescent protein, mKO2, conjugated with a DNA replication factor, Cdt1, which is expressed in the G1 phase and degraded in the S, G2, and M phases (measured via an FL2 channel) [33,34]. The daughter cells were colorless following cellular division since they did not produced either of those fluorescent proteins.

Both cell lines were cultivated in a chemically defined medium, TCX6D (Xell AG, Bielefeld, Germany), supplemented with 6 mM glutamine (Gln). To prevent the loss of the transgene antibody, 200 nM methotrexate (MTX, Sigma-Aldrich, St. Louis, MI, USA) was added during the pre-culture but it was omitted during batch cultivations. Cells were then incubated at 37 °C with 5% CO_2_ on an orbital shaker at 185 rpm with 50 mm deflection and subcultured every 2–3 days at a seeding density of roughly 3 × 10^5^ cells/mL. All cultivations were performed without antibiotics or any other supplements.

### 2.2. Stage Batch Cultivation in Shakers

Batch cultivations of both CHO DP-12 and FUCCI-transduced CHO DP-12 cells were performed in triplicate for each of the four osmotic conditions (300, 370, 460, and 530 mOsm/kg) in a working volume of 30 mL. The 300 mOsm/kg condition corresponds to physiological conditions and it is used as a control. Cells were seeded with a viable cell density (VCD) of 3 × 10^5^ cells/mL into a 125 mL polycarbonate un-baffled shake flask (Corning Life Sciences B.V., Amsterdam, The Netherlands) for the control (300 mOsm/kg) and the 370 mOsm/kg condition and at a VCD of about 6 × 10^5^ cells/mL for the 460 mOsm/kg and 530 mOsm/kg conditions as only insignificant cellular growth was expected in these conditions. The cells were then cultivated in batch modus as described above.

To create the osmolality stages, two different types of oversupplemented feeds (i.e., high-mannitol and high-glucose feed) were used. Both feeds were based on a CHO Basic Feed formulation (Xell AG, Bielefeld, Germany) containing 20 g/L (111 mMol) glucose and a subset of components already present in the growth medium. The high-glucose feed was supplemented with 404 mM glucose, 70 mM glutamine, and 27 mM asparagine. The high-mannitol feed, used as an osmotic control, was supplemented with 495mM mannitol and only 6 mM Gln to match the Gln concentration in the growth medium. The four osmolality stages were chosen to cover the full osmolality range tolerated by CHO DP-12 cells for at least 96 h without a viability drop under 80%. In this phase of cultivation, cells exhibit homeostatic behavior and proliferate at an optimum rate before the depletion of any essential nutrients [34]. For all four stages, 50% *v/v* of spent medium was used to exactly match the single-cell device cultivation conditions [35]. The medium-mixing protocol is shown in Table 1. The spent medium was prepared by centrifugation (300× *g*, 10 min) of the cell suspension of either CHO DP-12 or DP-12 FUCCI cells grown to a cell density of approximately 30 × 10^5^ cells/mL in an extra shaker under physiological conditions (300 mOsm/kg, 37 °C, 5% CO_2_).

This mixing protocol ensured high experimental reproducibility throughout all performed cultivations. Batch cultivations were closely monitored for 0–96 h before depletion of any essential nutrients to ensure an ≥80% viability in all conditions. Daily measurements included the following: VCD, viability, and cell diameter using a Cedex AS20 system (Innovatis-Roche AG, Bielefeld, Germany), as well as glucose and lactate concentration assessments (Biosen C-line Clinic (GEMAR GmbH, Celle, Germany). Osmolality was measured daily by a freezing point osmometer (Gonotec GmbH, Osmomat auto, Berlin, Germany).

### 2.3. Microfluidic Single-Cell Cultivation

The microfluidic cultivation was performed by applying a microfluidic cultivation device, as described previously [36]. This device was manually loaded using an untreated CHO DP-12 cell suspension cultivated at 300 mOsm/kg up to a VCD of 30 × 10^5^ cells/mL. The cells were compressed between the top and bottom surfaces of the cultivation chamber and had a rough cylindrical shape. Each array was constantly perfused with one of the four medium mixtures (300, 370, 460, and 530 mOsm/kg, see Table 1 for the medium mixing protocol) through supply channels. For this process, four syringes containing 22 mL of medium mix with the corresponding osmolality were connected to low-pressure syringe pumps (neMESYS, CETONI, Korbussen, Germany), each expelling 2 μL/min of the selected medium. These cultivation conditions were maintained via a microscope cage incubator and an additional CO_2_ incubation chamber (H201-K-FRAME GS35-M OKO touch, Okolab S.R.L., Pozzuoli, Italy) at 37 °C and 5% CO_2_. These conditions matched those for the shake flask stage batch cultivation. Microscopic images of the selected growth chambers were taken at intervals of 20 min with a 40× objective using an inverted phase-contrast microscope (Nikon Eclipse Ti2, NIS Elements AR 5.20.01 Software, Nikon Instruments, Amsterdam, The Netherlands).

### 2.4. Single-Cell Diameter Analysis

The microfluidic chambers with suitable final cell densities at 96 h were each numbered and then five were selected at random. Out of each selected chamber, one or two cells were randomly picked and processed by the same observer. Per condition, five cells were measured at a 1 h interval. Each cell was manually traced on enlarged microscopic images at its visible boundary and the area of the circle was calculated automatically. Since the size and height of the chamber are known, the tracked circle area was used to calculate the volume and the diameter of the spherical cell. The single-cell area was analyzed using an ImageJ 1.53 n [37]. The contouring of the cells was done using a GraphPad IntuosPro L (GraphPad, San Diego, CA, USA).

### 2.5. Mitochondrial Mass Assessment

A MitoTracker^®^ Green FM, (ThermoFisher Scientific, Waltham, MA, USA) mitochondrial dye (Cell Signalling Technology Inc., Danvers, MA, USA) was used to assess the mitochondrial mass inside the cell via flow cytometry. This fluorescent dye accumulates in mitochondria regardless of their membrane potential and, accordingly, it is commonly used for this purpose [38]. A total of 1.5 × 10^6^ cells were sampled from each biological replicate on day 2 and 4 (48 h and 96 h) after exposure to the hyperosmotic treatment. The cells were then centrifuged at 200× *g* for 5 min and resuspended in 750 μL of equiosmolar growth medium mix containing 150 nM of the dye. The staining concentration was adjusted to sufficiently resolve the stained and unstained populations measured via flow cytometry at the lowest voltage. Samples were incubated for 30 min at 37 °C in the dark on an orbital shaker. After incubation, the cells were counterstained with 1 µM 4′,6-diamidin-2-phenylindol (DAPI, 1:1000 dilution of 1 mM DAPI stock, Merck KGaA, Darmstadt, Germany), vortexed, and incubated for an additional 5 min at 37 °C in the dark. Cells were strained through nylon-mesh CellTrics filters (30 μm pore size) (Sysmex, Kōbe, Japan) directly into SuperClear^®^ cell culture tubes (Labcon, Petaluma, CA, USA), compatible with the automated carousel of the flow cytometer. Samples were measured via flow cytometry in a randomized order to minimize the bias due to the inherent fluorescence fading of the dye [39].

### 2.6. Mitochondrial Activity Assessment

A MitoTracker^®^ Red CMXRos (Cell Signalling Technology Inc., Danvers, MA, USA) staining was performed, as described in [11]. In brief, 1.5 × 10^6^ cells were stained in 750 µL of equiosmotic medium that contained either 100 nM of the dye or the growth medium as a vehicle control. The cells were then incubated for 30 min at 37 °C in the dark, counterstained with DAPI, and strained through cell mesh directly into a measuring tube, as described above.

### 2.7. Relative mtDNA Copy-Number Analysis via qPCR

#### 2.7.1. DNA Extraction

Genomic DNA was extracted from 2.0 × 10^6^ cells using the Wizard^®^ Genomic DNA Purification Kit (Promega, Madison, WI, USA), according to the manufacturer’s instructions. In the last step, the DNA was precipitated via an addition of isopropanol to the protein-free supernatant and desalted. The DNA was then washed with 70% ethanol to remove the isopropanol, and thereafter, dissolved in 50 µL of sterile MilliQ. The DNA solution contained 300–600 ng of DNA per μL, which was quantified photometrically using NanoDrop™ One (Thermo Fisher Scientific, Waltham, MA, USA).

#### 2.7.2. Primers

Primers were designed using the Primer-BLAST (National Center of Biotechnology Information, NCBI, Bethesda, MD, USA) primer design tool. For the target regions, we used a mitochondrial (mt) gene from a 16S ribosomal RNA (16S ribosomal RNA) (found in *Cricetulus griseus* mitochondrion, complete genome: NC_007936.1). The 16S rRNA is involved in mitochondrial protein biosynthesis [40,41]. The *B2m* (β2-microglobulin) gene was selected as a genomic reference as this gene does not seem to be involved in the hyperosmotic response (unpublished data based on proteomic analysis).

Primer pairs (shown in Table 2) were selected based on their specificity, thermodynamic properties (melting temperature, Tm), and non-self-complementarity. We also designed the nuclear and mitochondrial primer pairs to be suitable for simultaneous amplification. 

#### 2.7.3. qPCR Settings and Experiment

Quantitative PCR was performed using a LightCycler II 480 (Roche, Basel, Switzerland) detection System. The PCR recipe was as follows: 50 ng template DNA, 2× GoTaq^®^ qPCR Mastermix (Promega, Madison, WI, USA), 0.3 μM of each primer, and water added to the final volume of 20 μL. The PCR temperature cycling used was as follows: initial denaturation at 95 °C for 2 min, followed by 40 cycles of denaturation at 95 °C for 15 s, annealing at 60 °C for 30 s, and extension at 72 °C for 1 min. The melting curve was estimated after the amplification, utilizing a 50 → 97 °C at a 2.2 °C/s temperature gradient. Three to four technical replicates per biological replicate were measured. The samples were taken after 48 h and 96 h hyperosmolality exposure.

#### 2.7.4. Data Analysis

To estimate the relative copy number of the mitochondrial genome, CP values were first calculated using a LightCycler 480 intrinsic software (LightCycler^®^ 480 Software, version 1.5, Roche, Basel, Switzerland) via the 2nd derivative method.

To determine the relative gene copy number *R* of the target gene with respect to the reference gene, the efficiency-corrected quantification model by Pfaffl [42] (shown in Equation (1)) was used. It includes the efficiency E of the PCR reactions, which ideally is equal to 2, but efficiencies between 1.8 and 2.1 are acceptable [43].
(1)$$R=\frac{E_{Target}{}^{-{\bar{\;CP}}_{Target}}}{E_{Referenz}{}^{-{\bar{CP}}_{Referenz}}}$$

The relative gene copy number *R* of the target gene was normalized to the inter-run-control (IRC, a sample of a CHO-DP-12 pre-culture) using Equation (2). This compensated for possible differences between measurements on multiple plates.
(2)$$R_{norm}=\frac{R_{Target}}{R_{IRC}}$$

### 2.8. FUCCI-Transduced CHO DP-12 Cells-Based Cell Cycle Assessment

The effect of hyperosmolar exposure on the cell cycle was accessed using the unstained and untreated FUCCI-transduced DP-12 cells. Samples from each biological replicate were taken at 48 h and 96 h after hyperosmotic exposure. A total of 1.5 × 10^6^ cells were harvested and centrifuged at 200× *g* for 5 min. The samples were then resuspended in 750 µL of osmolality-adjusted growth medium mix to achieve an even cell density and transferred through mesh CellTrics filters (30 µm pore size (Sysmex, Kōbe, Japan)) into flow cytometry measurement tubes (Labcon, Petaluma, CA, USA).

### 2.9. Flow Cytometry

The prepared samples for each staining, acquired on a Navios EX flow cytometer (Beckman Coulter, Brea, CA, USA) equipped with a Kaluza v. 1.0 software and a standard filter setting, were randomized within the batch. Measurements were performed 48 h and 96 h following cellular exposure to osmotic stress. The protocols for the MitoTracker^®^ Green FM, (ThermoFisher Scientific, Waltham, MA, USA) and MitoTracker^®^ Red CMXRos (Cell Signalling Technology Inc., Danvers, MA, USA) Ros-stained cells, as well as for the intrinsically fluorescent CHO DP-12 FUCCI cells were optimized to plot the whole cell population in all osmotic conditions. The same parameters were used consistently for all measurements throughout all cultivations.

Data were analyzed using the FlowJo v. 10.7.1 (FlowJo, now BD Bioscience, Ashland, OR, USA). The hierarchical gating strategy included dead cell exclusion via a DAPI counterstain (for mitochondrial stains) and via a FCS area vs. SCS plot (for unstained DP-12 FUCCI cells). Doublets discrimination was performed based on the diagonal FCS area vs. FCS height plot. Subsequently, the median fluorescence in the corresponding channel (mitochondrial mass based on the MitoTracker^®^ Green FM, (ThermoFisher Scientific, Waltham, MA, USA)) (FL1 channel, 525/30 nm bandpass, BP) and the mitochondrial activity based on the MitoTracker^®^ Red CMXRos (Cell Signalling Technology Inc., Danvers, MA, USA) (FL2 channel, 575/30 nm BP)) were determined. For the DP-12 FUCCI cells, which expressed two fluorescent proteins according to the cell cycle phase, we used a gating of a sub-population protocol used by Pauklin, et al. [44]. It is based on an FL1 vs. FL2 (both 488 nm laser) plot which tracks the expression of mVenus (green, S/G2/M phase, measured via an FL1 channel) vs. mKO2 (red, G1 phase, measured via FL2 channel).

The same gating strategy was implied consequently throughout all measurements in high-glucose and high-mannitol cultivations.

### 2.10. Statistical Analysis

Data are presented as mean ± the standard error of the mean (*n* = 3). Statistical analysis was performed with an Excel Data-Add package (version 2016). Values of *p* < 0.05 were considered significant and the notations of * (*p* < 0.05), ** (*p* ≤ 0.01), and *** (*p* ≤ 0.001) were used for comparison with the control group or between the groups by Student’s t-test.

### 2.11. Growth Rate and Doubling Time (t_d_) Calculation

The growth rate was calculated using the following formula:$$\mu =\frac{\ln\left(x_{1}\right)-\ln\left(x_{0}\right)}{t_{1}-t_{0}}$$
where *μ* (h^−1^) is the growth rate, *x*_1_ (10^5^ cells/mL) is the viable cell density at time *t*_1_, *x*_0_ (10^5^ cells/mL) is the viable cell density at time *t_0_*. The doubling time *t_d_* (h) was calculated based on the growth rate as follows:$$t_{d}=\frac{\ln\left(2\right)}{\mu }$$

## 3. Results and Discussion

### 3.1. Influence of Hyperosmolality on Cell Growth

We first evaluated the effect of hyperosmolality on the viable cell density (VCD) and lactate accumulation in cultures grown in shake flasks. The cells were seeded from the same pre-culture directly into the medium mix of the corresponding osmolality. We used the following two supplements to increase the osmolality: (1) a nutrient-oversupplemented feed with glucose as the main reagent and (2) a mannitol-oversupplemented feed. A purposeful comparison of the effects on CHO cells between nutrient oversupplementation controlled by mannitol was lacking in the previous literature.

The relative VCD plots (shown in Figure 1A) for the CHO DP-12 and DP-12 FUCCI cells show a clear correlation of growth with osmotic pressure increase. Higher ambient osmolality causes lower VCD in both cell lines and for both added feed compositions. Figure 1A shows the relative increase of the VCD averaged for three replicates for each cultivation condition. The VCD was divided over the start VCD to exclude small discrepancies in initial seeding cell density. The original viable cell density profiles for both cultivations are included in Appendix A, Figure A1. In general, mannitol-exposed cultures displayed lesser growth depression compared with the glucose-treated ones. Thus, the addition of a mannitol-supplemented feed up to an osmolality of 370 mOsm/kg does not seem to affect the cellular proliferation in either CHO DP-12 cells (shown in Figure 1A, left) or DP-12 FUCCI cells (shown in Figure 1A, right). Conversely, when the cells were exposed to a high-glucose feed at the same osmolality (370 mOsm/kg), we registered a significantly lower cell density. A very similar effect on cellular growth has been reported in a recent study, which also used an oversupplemented feed for osmolality increase [12]; here, the suspension-grown CHO cells also showed no cellular growth at 500 mOsm/kg.

We punctually checked the influence of the hyperosmolar treatment on cellular productivity. We have already reported that the cell-specific production rate (qP) was not influenced by the exposure of DP-12 cells to the oversupplemented feed [11]. For the current protocol, we measured the qP for the 460 mOsm/kg osmolality stage for high-glucose (Glc), high-mannitol (Man), and control conditions. The measurement was performed on day four. The cell-specific antibody production rates were stable in all three conditions and were similar to those reported in the aforementioned publication. The rates were as follows: qP_Glc_ was 9.12 ± 0.35 pg/cell × day, qP_Man_ 8.94 ± 0.52 pg/cell × day, and qP_control_ 9.52 ± 0.65 pg/cell × day.

Next, we evaluated the influence of high-glucose and high-mannitol feed exposure on the doubling time *t_d_* (shown in Table 3). From the data, we can conclude that both reagents caused a significant stepwise increase in the doubling times of the cells with a non-linear effect on cellular growth (shown in Table 3, Appendix A, Figure A2). Growth depression tends to be more pronounced for high-glucose-exposed cells, although the available data does not allow to precisely quantify this effect.

The osmotic stages (shown in Appendix A, Figure A3A) exhibited highly comparable profiles between both the high-mannitol and high-glucose feed cultivations. The only exception was a slight discrepancy of 30 mOsm/kg (average for days 0–4: 435 mOsm/kg for glucose-supplemented feed and 465 mOsm/kg for mannitol-supplemented feed) in DP-12 FUCCI cells stage cultivations. Otherwise, the stages were achieved with a high accuracy for each cell line. Therefore, we conclude that the differences we observed in growth, viability, and diameter between the mannitol- and glucose-exposed cells were not caused by differences in the ambient osmolality.

Glucose concentration profiles are shown in Appendix A, Figure A3B. In mannitol-supplemented cultures, glucose concentration remained rather stable and did not exceed 55 mMol (day 2, 530 Man CHO DP-12 FUCCI). For glucose-supplemented stages, glucose concentration was much more stratified and reached 176.05 mMol on day 3 in the 530 mOsm/kg Glc DP-12 FUCCI and 215.3 mMol on day 1 for the CHO DP-12 530 mOsm/kg Glc. Glucose oversupplementation in concentrations above 100 mMol seems to have specific detrimental effects on the cells exceeding those caused by mannitol-induced osmotic effects, as it can be seen from the VCD plot in Figure 1. The measured lactate concentrations, shown in Appendix A, Figure A3, did not exceed 20 mMol and are known to be well tolerated by the cells [45].

The cells were highly viable in all cultivations, with viability of ≥95% at seeding. Viability remained within an interval of 90–100% for all measurements in all cultivations for both cell lines, except for the stage with the highest osmotic pressure of 530 mOsm/kg where viability dropped gradually from about 95 on day 0 to 83.2% (DP-12 FUCCI, high-mannitol feed), 77.1 (DP-12 FUCCI, high-glucose feed), 93 (CHO DP-12, high-mannitol feed) and 83.2% (CHO DP-12, high-glucose feed) on day 4 (Figure A1). Generally, exposure to a high-glucose feed leads to a more significant viability drop. Similar conclusion was achieved via comparison of the effect of acute hyperglycemia (100 mMol glucose) and mannitol (100 mMol mannitol) exposure on cultured pheochromocytoma (PC12) cells [46]. A more pronounced viability drop in the FUCCI culture suggests that the incorporation of a FUCCI sensor might have rendered the DP-12 FUCCI cells less tolerant to osmotically induced stress than the parental CHO DP-12 cell line. Stable viability profiles achieved in all cultivations also ensured that subsequent analyses, especially those where dead cells were not specifically excluded (i.e., mtDNA qPCR analysis), returned reliable results. In the presence of a neutral osmotic reagent (mannitol), even the harshest osmolality condition (530 mOsm/kg) resulted in a slow VCD progression during the observation time. Here, the initial VCD increased 2.69-fold between days 0–4 for the CHO DP-12 cells and 3.58-fold for the DP-12 FUCCI cells. In contrast, the VCD increased only marginally in the presence of a high-glucose feed in the 530 mOsm/kg stage. A 1.08-fold (CHO DP-12) and a 1.32-fold (DP-12 FUCCI) initial VCD increase was registered between days 0–4. These findings agree with several independent studies when employing only one osmotic reagent [9,11,13,23,26,47]. The untreated cells in the 300 mOsm/kg condition showed a similar VCD increase in all four cultivations. The CHO DP-12 cells showed a 23.05-fold (high-glucose cultivation) and a 22.45-fold (high-mannitol cultivation) initial VCD increase during the 0–4 days observation period. The DP-12 FUCCI cells showed a 26.41-fold increase during the high-mannitol cultivation and a 22.42-fold VCD increase in the high-glucose cultivation. The significant discrepancies of achieved viable cell densities between osmotic stages of 300–530 mOsm/kg did not result in likewise different cell viabilities. However, our observations not only agree with previous findings, but also reinforce the conclusion that high-glucose feeds exhibit negative effects on cellular growth and viability, exceeding those of osmotic effects, as growth rate and viability depression were much more pronounced for both cell lines in high-glucose conditions when compared to the high-mannitol exposures.

### 3.2. Influence of Hyperosmolality on Cell Diameter

#### 3.2.1. Populational Observation of Cell Diameter Dynamics

The occurrence of enlarged cells upon exposure to osmolality has been previously documented in CHO cells, including in our research [11,13]. Upon exposure to hyperosmolality, which causes an efflux of water from the cytoplasm, the cells strive to regain their initial volume to prevent macromolecular crowding inside the cell. The first hours of adaptation are possibly guided by an onset of a regulatory volume increase [48,49,50]. This process involves swelling of osmotically shrunken cells by an exchange of inorganic osmolytes (such as Na^+^ and Cl^−^) with organic osmolytes (such as betaine and taurine) in the cytoplasm. However, such a response does not explain the occurrence of extremely oversized cells which also retain their significantly greater mass upon drying [14], suggesting that these cells actually do accumulate biomass and not merely osmolytes.

Moreover, most published studies investigating CHO cells have either used only a single reagent to increase osmolality without including any sort of osmotic control [13,51,52,53] or otherwise deployed a control reagent which has its own specific known effects on the cells [12]. As a result, the available existing data are inconsistent and do not allow us to disentangle the effects of hyperosmolality and the specific effects of the added reagents. In our setup, we used an industrially relevant feed as a basis and then supplemented it with nutrients (glucose, asparagine, and glutamine, as already discussed in the Materials and Methods section) or with mannitol only to create two supplemented feeds (i.e., high-glucose and high-mannitol) of the same osmolality. While it is theoretically possible that mannitol may have yet unknown specific inhibitory effects on cellular growth, a review of the existing literature establishes that it is by far the best osmotic control, at least according to currently available data [30].

As shown in Figure 1A, cells grow at a maximum rate during physiological osmolality. The mean diameter of the population at 300 mOsm/kg was narrowly maintained between 13.8 ± 0.14 µm (CHO DP-12) and 14.9 ± 0.14 µm (DP-12 FUCCI) throughout days 0–4 (Figure 1B,C). In this condition, almost all cells were wired to complete a doubling cycle. Elevation of the ambient osmolality causes an increase in the mean diameter and volume (Figure 1D). The most significant diameter or volume shift was seen within the first 24 h of the treatment, but remarkably, the volume gain was retained only in the 530 mOsm/kg in the high-glucose and high-mannitol conditions for both cell lines past the 24 h mark of exposure. We observed the maximal relative diameter (143 ± 9% compared to the control cells) and volume in the 530 mOsm/kg osmotic stage for the high-mannitol and high-glucose treatment (shown in Figure 1D). For the 460 mOsm/kg stage, the relative diameter gain (122 ± 4%, relative volume 180 ± 13%, compared to control cells) matches almost exactly the diameter gain of a different CHO strain (about 125% compared to the control) exposed to hyperosmolar NaCl (490 mOsm/kg) [13]. It is also important to notice that the cells in the 370 mOsm/kg Man condition exhibited a very small decrease in VCD and doubling time but gained significantly in their mean diameter (shown in Figure 1A–C).

As it has been shown in previous studies and it was described above, the mean diameter of the cell population positively correlates with the ambient osmolality increase. This has previously led researchers to conclude that there is a direct dose-dependent effect between the two factors, i.e., that higher osmotic pressure leads to a faster and more pronounced increase in cellular diameter (shown in Figure 2, left). However, population-wide observations can mask the behavior of single cells. In other words, during heterogeneous cellular stress response, some cells manage to retain their normal size and continue to proliferate, but other cells get engaged in mass gain without proliferation (shown in Figure 2, right). This could lead to the same observed mean population-level outcomes. In that case, a stepwise increase in osmolality would shift the fraction of the cells that are still able to proliferate and those that cannot (moderate and intense osmolality increase, shown in Figure 2).

To understand the exact mechanism of the cell volume gain in response to hyperosmolality exposure, we, therefore, purposely set out to analyze the dynamics of single cells within a microfluidic cultivation device [36].

#### 3.2.2. Single-Cell Analysis Reveals Heterogeneity in Response to Hyperosmolality

We used a microfluidic single-cell cultivation device to directly measure the growth and diameter dynamics of single cells. We were especially interested in cells grown in the 370 and 460 mOsm/kg stages, as we expected either a rather homogenous pattern of cellular diameter increase (see Figure 2, left) or a more differentiated one (see Figure 2, right). We expected all cells to engage in proliferation under normal conditions (300 mOsm/kg) and almost all cells to gain mass, but not divide, under 530 mOsm/kg.

The results of the cellular diameter measurements are shown in Figure 3. Cellular division is seen as a vertical drop in the diameter curve. The calculated t_d_ for each condition is listed below the graphs in Figure 3A,B. Although the doubling time estimation was performed only on five cells, the results still correlate well with the population data, which are presented in Table 3. The cells under the normal cultivation conditions (300 mOsm/kg) divide four to five times within the observation period (96 h). The cells in the 370 Glc and the 460 mOsm/kg Glc and Man conditions show mixed population behavior which strongly supports the suggestion of the mechanism shown in Figure 2, right, i.e., that cells either divide (although at a slower rate) and retain their normal diameter range or accumulate mass leading to an overall observable diameter increase.

Even at 530 mOsm/kg, rare cells do occasionally manage to complete a few divisions. Only one division was spotted in the 530 mOsm/kg Glc condition, whereas two dividing cells with a total of four divisions were observed in the 530 mOsm/kg Man condition. Most of the cells, however, completely lose their ability to complete a successful cell division (see Figure 3A,B, right) and become significantly and visibly larger in size (see Figure 4A).

We, therefore, conclude that, on the single-cell level, CHO DP-12 cells respond rather heterogeneously to oversupplemented feed exposure (see Figure 3). By increasing the ambient osmolality, some cells are forced to exit the normal proliferation pattern, but they remain metabolically active and continue to accumulate biomass. Cells respond heterogeneously to the treatment, where some cells already abort proliferation at 370 mOsm/kg, whilst others manage to sustain divisions (albeit at a much slower rate) even under the highest osmolality of 530 mOsm/kg.

Finally, we sought to evaluate how mass accumulation changes in response to the ambient osmolality increase. If we only consider the population level observations of the cell diameter (see Figure 1B–D), we might conclude that the mass accumulation accelerates proportionally to the hyperosmotic pressure. However, based on the single analyzed cells (see Figure 3), we calculated the rates at which the cell diameter increases. This rate (see Table A1) corresponds to the slope of the linear fit function (see Figure 4B) based on the relative diameter data of a single randomly chosen cell in each condition. As the slope of the linear fit function decreases (see Table A1, Figure 4B), the cell diameter gain rate progressively slows down as osmolality rises. Therefore, the individual cells of the untreated population gain mass at a higher rate compared to the high-mannitol and high-glucose feed exposure.

The mass gain rate decreases 10-fold between the physiological condition and the high-glucose feed at 530 mOsm/kg and 3.3-fold between the 300 and the 530 mOsm/kg when exposed to a high-mannitol feed. We, therefore, conclude that an observed increase in the average diameter of the cell population is due to the proportion of the cells which are engaged in the mass accumulation, rather due to any division increases concomitantly with ambient osmolality. Furthermore, individual cells tend to accumulate mass slower as the ambient osmolality increases (see Figure 4B).

### 3.3. Influence of Hyperosmolality on Mitochondria

Some aspects of the role of mitochondria under hyperosmotic stress have already been elucidated in several independent studies [8,11,18,19,20,21], but consistent observations on mitochondrial density and activity in CHO cells are sorely lacking in the literature. In this current study, we set out to measure mitochondrial mass, membrane potential (ΔΨm), and mtDNA copy number in the cell.

Following mitochondrial assays were performed on the samples taken from the same culture: qPCR analysis of the relative mtDNA copy number and fluorescent mitochondrial staining, which correlate with the mitochondrial mass (MitoTracker^®^ Green FM, (ThermoFisher Scientific, Waltham, MA, USA)) and the bulk mitochondrial membrane potential (MitoTracker^®^ Red CMXRos (Cell Signalling Technology Inc., Danvers, MA, USA)) in the cell. MitoTracker dyes accumulate in the mitochondrial matrix and bind covalently to free thiol groups of a subset of mitochondrial proteins [54]. These assays were all performed with CHO DP-12 cells from the same cultivations taken 48 h (day 2) and 96 h (day 4) after hyperosmolar exposure. With these analyses, we also aimed to refine and confirm our previous finding that high-glucose feed-exposed cells exhibit significantly increased mitochondrial membrane potential in a similar set-up [11].

When exposed to a high-glucose feed, the CHO cells seem to react with an increase in mtDNA copy number, mitochondrial mass, and mitochondrial membrane potential (shown in Figure 5A,B), which increase at the same rate as the cell diameter only on day 2 (see Figure 5D, left).

The increase in mitochondrial fluorescence and mtDNA copy number, on a populational level, positively correlated with the stage-wise increase in ambient osmolality on day 2, but the measured mitochondrial membrane potential and mass dropped significantly on day 4, compared to the physiological condition (see Figure 5D). The mtDNA copy number retained the same stage-wise increase as it was observed on day 2. Such observations may suggest a shift in mitochondrial functionality occurring between the first and the second measurement.

At first glance, high-glucose feed-exposed cells appear to exhibit the same high density of mitochondria as normal untreated cells on day 2 (see Figure 5D). Mitochondrial quantity changes with the proliferative capacity of the cell [55]. Throughout the batch cultivation, mitochondrial density is at the highest in the exponential growth phase and then declines towards the stationary or quiescent phase as proliferation ceases. Therefore, the occurrence of the same mitochondrial density in high-glucose-exposed cells, whose proliferation is substantially impaired, would be a highly unusual and counterintuitive result. Additionally, there seemed to be an underlying cause of a mitochondrial fluorescence drop in the 370–530 mOsm/kg high-glucose stages compared to the control on day 4 (see Figure 5A, Glc feed); we observed such a pattern at least three times for biologically independent cultivations following the same protocol. We, therefore, considered that there might be a confounding factor that could contribute to the observed increase in the mitochondrial membrane potential, mass, and relative mtDNA copy number as the ambient osmolality surges.

To explore this possibility, we focused on the connection between high-glucose and hyperosmolar exposure and their relation to cellular stress. Exposure to high-glucose concentrations, as well as to severe mannitol-induced hyperosmolality, can serve as a trigger to an inflammatory response in a cell [56,57,58]. In CHO cells, the high-glucose treatment causes the mitochondria to become “leaky” and expel reactive oxygen species (ROS), which propagate inflammation and cellular stress [16,59,60,61,62]. In turn, ROS accumulation causes an initial increase in the mitochondrial membrane potential [63,64] and, thus, it is responsible for an observed increase in the mitochondrial membrane potential-sensitive MitoTracker^®^ Red CMXRos (Cell Signalling Technology Inc., Danvers, MA, USA) fluorescence. Following an initial period of hyperpolarization, the mitochondrial membrane will eventually depolarize if the exposure persists [62], which could potentially explain the observed drop of the observed MitoTracker^®^ Red CMXRos (Cell Signalling Technology Inc., Danvers, MA, USA) fluorescence on day 4. The mitochondrial membrane depolarization (see Figure 5D) on day 4 in the 460 mOsm/kg and 530 mOsm/kg high-glucose conditions, as evidenced by its drop below basic level in the untreated cells, most likely leads to a reduced ATP production and basal respiration per single mitochondrion, as well as an overall shift towards a glycolytic metabolic profile [18,59,65], which also correlates with our data on increased lactate production (see Figure A3). Diminished mitochondrial biogenesis, decreased mitochondrial number, and altered membrane potential caused by a prolonged hyperglycemia (registered in our data by a drop of fluorescence of both mitochondrial dyes on day 4 as compared to day 2) may cause mitochondrial dysfunction [66], mitochondrial swelling, and, in extreme cases, ruptures of the mitochondrial outer membrane [62,67,68]. We can therefore conclude that exposure to a high-glucose feed leads to a dose-dependent increase of oxidative stress in CHO cells which, in extreme cases (460 and 530 mOsm/kg high-glucose), can lead to mitochondrial dysfunction, which, in turn, is a key modulator of premature stress-induced senescence [69].

Oxidative stress also can influence the staining with the mitochondrial mass-dependent MitoTracker^®^ Green FM, (ThermoFisher Scientific, Waltham, MA, USA) fluorescent marker, although its accumulation has previously been found not to be influenced by the mitochondrial membrane potential [70]; oxidative stress and cytosolic and mitochondrial ROS production significantly correlate with its concentration in mitochondria and, therefore, with the measured fluorescence [71]. According to some studies, MitoTracker^®^ Green FM, (ThermoFisher Scientific, Waltham, MA, USA) accumulation can also be influenced by the ΔΨm [72]. The measured increase in mitochondrial fluorescence is, therefore, a composite of possibly increased mitochondrial mass and the response of mitochondria to glucose-induced oxidative stress.

Further, ROS-induced mtDNA damage causes relaxation of the mtDNA supercoiled conformation, which, in turn, facilitates the accessibility of the sequences with minimal damage to DNA polymerase and leads to a decrease in the threshold cycle (Ct) value [73,74]. Therefore, the measured increase in the relative mtDNA copy number (see Figure 5B) also reflects the combined effect of the true copy number increase per cell and of the mtDNA microdamage accumulation, which, as expected, positively correlates with the ambient osmolality and reaches its peak in the 530 mOsm/kg high-glucose condition.

The introduction of an osmotic control into the experiment helped to disentangle these combined effects. In the high-mannitol-exposed cells, we observed a different cellular phenotype regarding mitochondrial parameters. As it can be seen in Figure 5A,C, exposure to a high-mannitol feed (370–460 mOsm/kg) triggered no significant increase in the mitochondrial parameters, relative to the untreated population on both measurement time points. This produced an apparent dilution of mitochondria within the increasing volume of the cell, as it is clearly shown in Figure 5D. It is interesting to note that the highest mannitol-induced osmolality of 530 mOsm/kg does induce a statistically significant rise in mtDNA copy number, mitochondrial membrane potential, and mitochondrial mass (although they do not scale with the cellular volume, see Figure 5A,C,D). This finding might point out a critical threshold of the cellular response to osmotic treatment lying between 460 and 530 mOsm/kg. Although a similar deflection point of the high-osmolality response of the CHO cells has been previously reported in the literature [18,75], the exact reasons for such behavior remain unclear.

As a result, we can conclude that in osmotically and high-glucose-stressed cells alike, mitochondria tend to become more diluted with the increasing cell volume. However, while a high-glucose feed, but not a high-mannitol feed, caused a dose-dependent degree of oxidative stress, the mitochondria responded to it with an initial membrane potential increase, followed by a subsequent drop, as well as an accumulation of mtDNA damage.

### 3.4. Cells Accumulate in the Late G1 Phase without Entering Mitosis upon Oversupplemented Feed Exposure

We analyzed the cell cycle distribution in DP-12 FUCCI cells by flow cytometry on days 2 and 4 of the batch culture. Both time points are nested in the exponential growth phase, as cells slow the growth speed only around day 6 of the typical batch cultivation of the untreated cells (see Appendix A, Figure A4).

We began by comparing the distribution of the cell cycle phases under normal physiological conditions. Reference cultivations with no added reagents in the glucose (top row) and mannitol (bottom row) stage experiments are shown in green in Figure 6. For both experiments, the cell cycle phase distribution of the untreated cells is remarkably similar, with only minimal deviations of less than 3% between the same phases measured at the same time points. Therefore, we presume that both the biological variability of the cells and any possible technical biases are negligible and the experiments can be compared to each other.

In the treatment conditions (370, 460 and 530 mOsm/kg, Glc and Man, respectively), we observed that exposure to both oversupplemented feeds leads to a dose-dependent accumulation of the cells in the late G1 phase. This effect was certainly more pronounced for the high-glucose feed-exposed cells than for those treated with mannitol, as well as the effect also became more pronounced with the increase of hyperosmotic pressure and with the exposure duration from day 2 to 4 (see Figure 6). This result confirmed the observations based on DNA staining published previously [11,14]. Simultaneously, the number of cells entering an early G1 phase and those with double DNA content in the S/G2/M phases was significantly depleted. The cells which contain the doubled DNA just before cell division are roughly doubled in size [76]. We also observed this “normal” doubling of the cell size on the single-cell level (see Figure 3). Because the proportion of such double-sized cells in the S/G2/M cell cycle phase was depleted proportionally to the increase in osmotic pressure in both treatments (see Figure 6), we can rule out that the mean populational (see Figure 1B–D) and single-cell volume increase (see Figure 3) can be explained by their accumulation.

Permanent cell cycle arrest and large cell size (discussed in Section 3.2) are both hallmarks of senescence [76,77]. As to the widely known replicative senescence characterized by irreversible proliferation abortion after a certain number of divisions, premature cellular senescence can also be induced by mitogens or oxidative stress [57,78]. The latter can directly induce cellular senescence [79,80] and produce non-replicative and enlarged senescent cells. Senescent cells are stably arrested with a G1 DNA content and lose their ability to enter the S phase [81], which also corresponds to our observations of the cell cycle under osmotic pressure.

Taken together, the cell cycle phase distribution of hyperosmolality-exposed cells strongly suggests that the transition of the primary cell cycle control checkpoint at the initiation of DNA synthesis (G1/S transition) [82] is impaired. This also supports the finding that the number of mitochondria in both osmotically challenged and high-glucose-exposed CHO cells does not seem to increase with the cellular volume (see Section 3.3), as mtDNA replication occurs predominantly during the S phase [83]. DNA doubling initiation appears to be the main hurdle for cellular division and cell cycle completion under severe mannitol-induced hyperosmolality and high-glucose stress. Together with increased cell size and mitochondrial impairment, cell accumulation before the S phase points to the onset of premature stress-induced senescence.

## 4. Conclusions

The current study demonstrates that exposure of suspension-grown antibody-producing CHO DP-12 cells to a stage-wise increasing osmotic pressure, mediated either by a relatively neutral high-mannitol feed or by a highly supplemented high-glucose feed, causes extensive changes in the cellular morphology, size, and proliferation of those cells. We registered a dose-dependent decrease in the proliferation rate, which was especially pronounced for the high-glucose conditions, and a concomitant increase in the average cell diameter.

By utilizing a single-cell cultivation platform, we were able to track individual cells and reveal that exposure to both oversupplemented feeds leads to a heterogeneous cellular response to hyperosmolality. The cells either retain their normal size and continue proliferation (albeit at a reduced rate) or they become deregulated and gain excessive mass, almost tripling in volume. The number of the tracked cells per condition (five) is sufficient for the principle validation of the heterogeneity hypothesis. A more thorough investigation aiming at a detailed quantification of these heterogeneities within the population will require an analysis of significantly larger cell numbers. This question offers a direction for future research.

Through the non-invasive measurement of the cell cycle-specific fluorescence of FUCCI-transduced cells, we were also able to document a clear accumulation of the cells in the late G1 phase, before the S phase. Our previous research also indicated no increase in DNA content upon hyperosmolar exposure [11]; therefore, it seems that both osmotic stress and high-glucose-induced stress impair DNA synthesis and hinder the cells from entering the S phase, posing a permanent cell cycle block upon the G1/S checkpoint. Here, again, glucose overfeeding induces a more pronounced effect.

Next, analysis of the mitochondrial membrane potential-sensitive (MitoTracker^®^ Red CMXRos (Cell Signalling Technology Inc., Danvers, MA, USA)) fluorescence stain, mass-sensitive (MitoTracker^®^ Green FM, (ThermoFisher Scientific, Waltham, MA, USA)) fluorescence stain, and mtDNA copy numbers shows that neither neutral hyperosmolar mannitol nor the high-glucose exposure induce new mitochondria generation; therefore, mitochondria must be diluted within the volume of enlarged cells. As the cells at higher osmotic conditions (both mannitol and high-glucose) almost triplicate their volume, the mitochondria are diluted and the DNA: Cytoplasm ratio is decreased. Additionally, ROS expelled under a high-glucose feed damage the mtDNA and induce depolarization of the mitochondrial membrane, especially in the 460 and the 530 mOsm/kg conditions. Large cell size, mitochondrial depolarization, cell cycle arrest, and decreased DNA:cytoplasm ratio are all hallmarks of premature stress-induced senescence which occurs in CHO cells upon sustained exposure to oversupplemented feed at 460 and 530 mOsm/kg. Senescence induction should be possibly omitted under industrial processes; therefore, osmolality and nutrient availability should be kept within physiological optimum.

Certain limitations should be noted concerning the current study. Although we found a number of morphological changes pointing towards the onset of premature stress-induced senescence in the CHO DP-12 cells under high-glucose feed exposure, we did not investigate direct markers (such as activation of senescence-associated ß-galactosidase or the p16-pRB pathway), which are activated in stress-induced senescence response [57,84,85]. Further research is needed to confirm this. However, considering the current results and previous knowledge, we suppose that high-glucose-induced oxidative stress, not osmolality alone, likely causes the excessive generation of reactive oxygen species in mitochondria and, thereby, accelerates stress-induced senescence in suspension grown CHO DP-12 cells. With these findings, our work represents the fundament for future investigations of bioprocess-relevant research questions in the field of stress-induced cell-to-cell heterogeneity.

## Figures and Tables

**Figure 1 cells-11-01763-f001:**
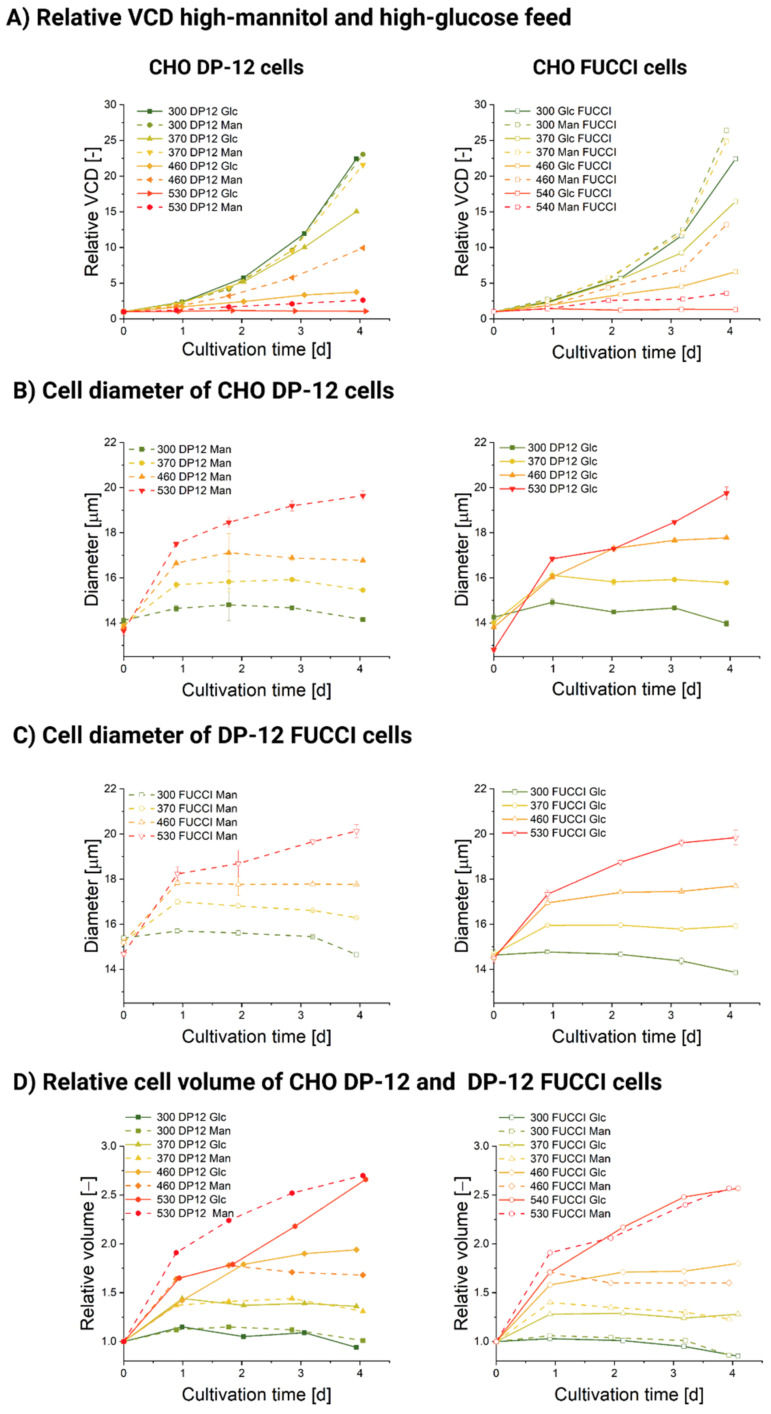
Cultivation data for stage cultivations of CHO DP-12 and DP-12 FUCCI cells. (**A**) Relative viable cell density (VCD) of CHO DP-12 and DP-12 FUCCI cells during stage cultivation under 300, 370, 460 and 530 mOsm/kg controlled either by addition of a highly supplemented feed (Glc) or the same feed, where only mannitol was added as control (Man). The cell densities were divided over the initial cell density to avoid the influence of small discrepancies in initial cell density. (**B**) Cell diameter of CHO DP-12 cells during stage cultivation with added high-supplemented feed (Glc) or high-mannitol feed (Man). (**C**) Cell diameter of DP-12 FUCCI cells during stage cultivation with added high-supplemented feed (Glc) or high-mannitol feed (Man). (**D**) Relative cell volume of CHO DP-12 and DP-12 FUCCI cells during stage cultivation with high-supplemented feed (Glc) or high-mannitol feed (Man).

**Figure 2 cells-11-01763-f002:**
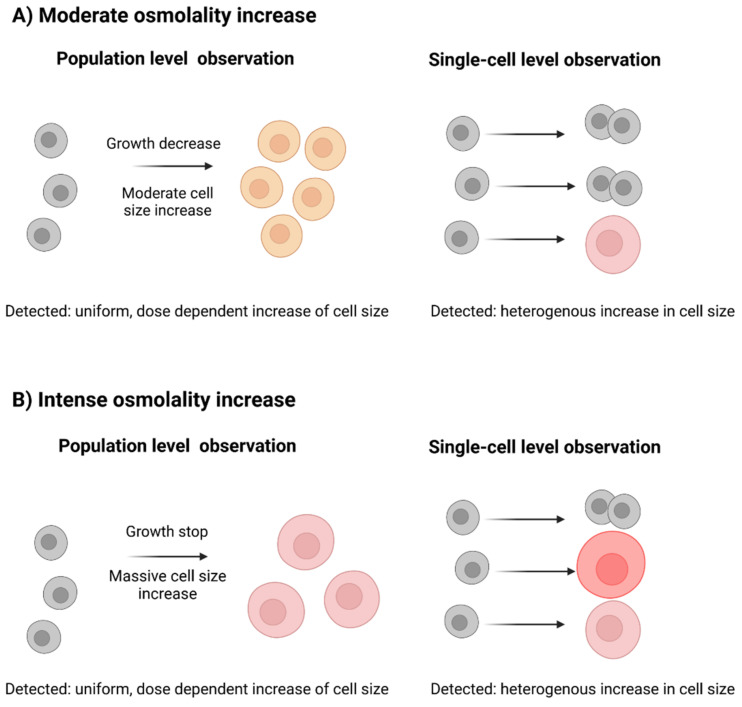
Cellular stress response, observed on a single cell or on a population level. (**A**) During a moderate osmolality increase, population-level observations register a uniform dose-dependent increase in cell size caused by hyperosmotic stress (left). However, the underlying heterogeneous increase in cell size can be observed only in the single-cell level (right). (**B**) During an intense osmolality increase, a more pronounced homogenous increase in cell size can be registered on a population level (left) or a heterogeneous cellular response can be seen on a single-cell level (right).

**Figure 3 cells-11-01763-f003:**
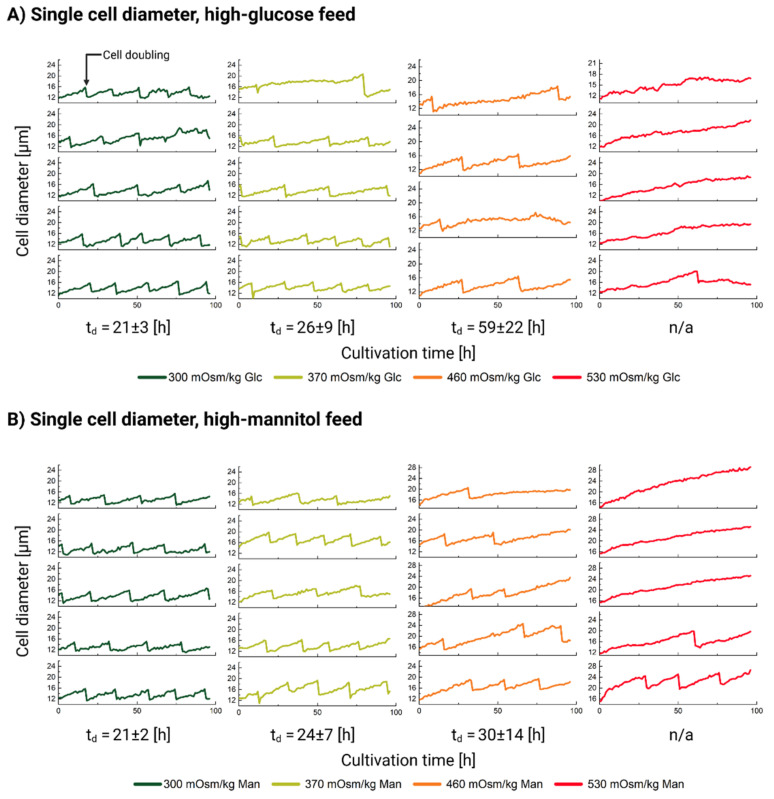
Cell diameter dynamics for 0–4 days (0–96 h) of the single-cell cultivation of CHO DP-12 cells exposed to high-glucose (panel (**A**)) and high-mannitol (panel (**B**)) feed. The stages are ordered from left to right: 300 mOsm/kg (left, green); 370 mOsm/kg (light green, second from left); 460 mOsm/kg (orange, third from left); 530 mOsm/kg (red, fourth from left). The window was chosen between either 10–26 µm or 14–30 µm to best fit the data. Cellular division is seen as a vertical drop in the diameter curve.

**Figure 4 cells-11-01763-f004:**
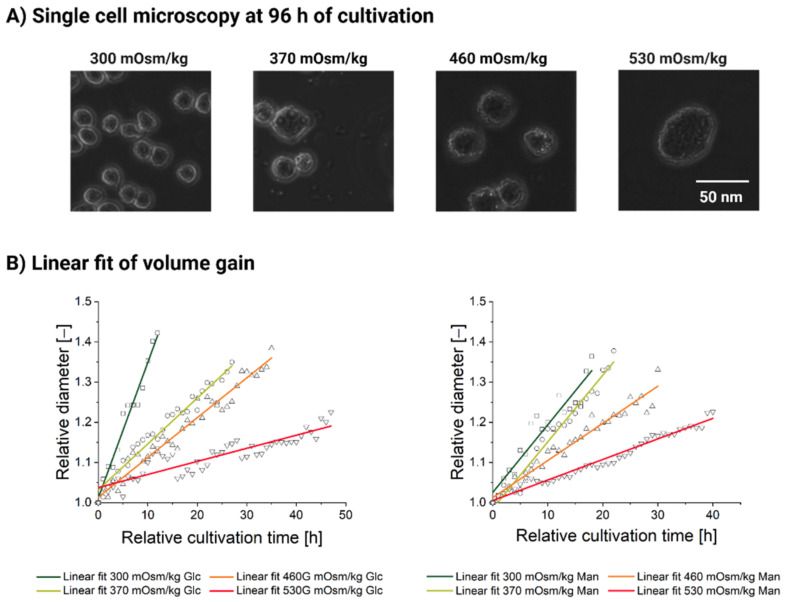
Data on single-cell cultivation of CHO DP-12 cells exposed to high-glucose and high-mannitol feed over the cultivation duration [h]. (**A**) Representative microscopic photographs of the CHO-DP-12 cells exposed for 96 h to different osmotic treatments (high-mannitol feed). (**B**) Relative diameter gain based on one randomly picked cell taken for one division period after at least 24 h of hyperosmolality exposure. Relative diameter [–], without dimension.

**Figure 5 cells-11-01763-f005:**
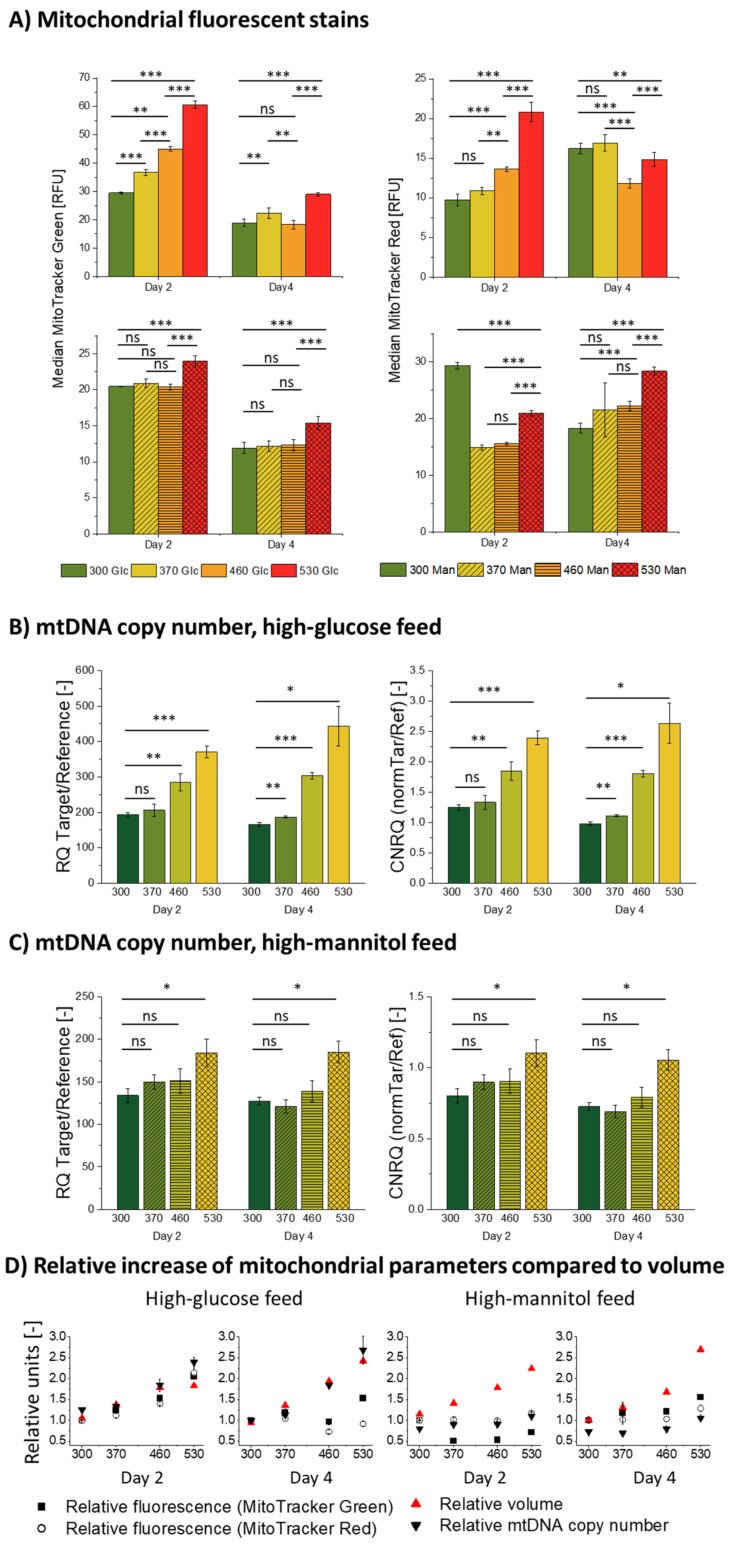
Mitochondrial parameters measured on day 2 and day 4 of the stage cultivations. Here, 300, 370, 460, and 530 stand for 300, 370, 460 and 530 mOsm/kg ambient osmolalities. (**A**) Median fluorescence intensities of the cell population previously gated for live and single cells of CHO DP-12 cells stained with MitoTracker^®^ Green FM, (ThermoFisher Scientific, Waltham, MA, USA) and MitoTracker^®^ Red CMXRos (Cell Signalling Technology Inc., Danvers, MA, USA) exposed to high-mannitol and high-glucose feeds determined by the flow cytometry analysis. Relative mtDNA copy numbers (RQ Target/Reference) and calibrated normalized relative quantities (CNRQ normTar/Ref) of the CHO DP-12 cells exposed to (**B**) high-glucose feed and to (**C**) high-mannitol feed. (**D**) Volume, mtDNA copy number, and median fluorescence intensities of MitoTracker^®^ Green FM, (ThermoFisher Scientific, Waltham, MA, USA) and MitoTracker^®^ Red FM-stained (Cell Signalling Technology Inc., Danvers, MA, USA) populations relative to the parameters measured in the untreated populations (300 mOsm/kg condition). Statistical significance by a two-tailed Student’s t-test with the statistically significant threshold of *p* < 0.05; the notations of * (*p* < 0.05), ** (*p* ≤ 0.01), and *** (*p* ≤ 0.001) were used.

**Figure 6 cells-11-01763-f006:**
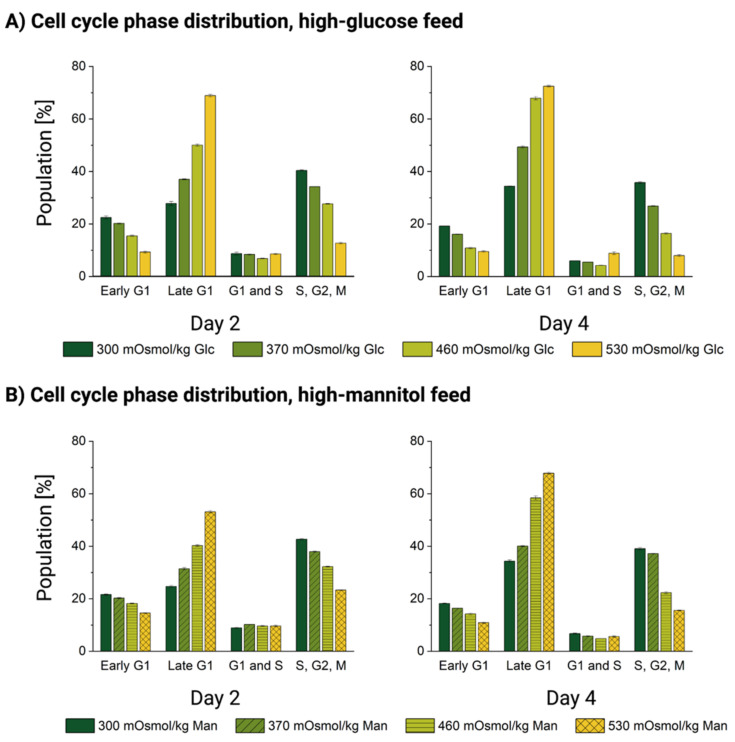
Cell cycle distribution [%] of DP-12 FUCCI cells exposed to high-glucose or high-mannitol feed. Dead cells and cell doublets were gated out based on the FSC/SSC and the diagonal FCS-A vs. FCS-H plots, described in the Materials and Methods section. The gating of the phases was performed uniformly throughout all analyses based on [44]. Cell cycle phase distribution [%] of the DP-12 FUCCI cells exposed to (**A**) high-glucose feed and (**B**) high-mannitol feed.

**Table 1 cells-11-01763-t001:** Mixing protocol of the feed and growth medium to reach the corresponding osmolality.

Condition	Spent Medium *	TCX6D w/6 mM Gln	80% High-Glucose or High-Mannitol Feed + 20% TCX6D w/6 mM Gln
300 mOsm/kg (control) **	50% *v*/*v*	50% *v*/*v*	-
370 mOsm/kg	50% *v*/*v*	16.65% *v*/*v*	33.3% *v*/*v*
460 mOsm/kg	50% *v*/*v*	33.3% *v*/*v*	16.65% *v*/*v*
530 mOsm/kg	50% *v*/*v*	-	50% *v*/*v*

* spent medium was achieved by centrifugation (300× *g*, 10 min) from an extra shaker, cultured until the cell density was approximately 30 × 10^5^ cells/mL. ** condition designations, e.g., 300 mOsm/kg or 370 mOsm/kg, are based on approximate values of the calculated mean osmolalities for all four cultivations—CHO DP-12 and DP-12 FUCCI cells, high-mannitol and high-glucose cultivation. The exact profiles of the measured osmolalities can be seen in the Figure A3.

**Table 2 cells-11-01763-t002:** Primer sequences used to amplify nuclear and mitochondrial DNA sequences. Forward (for) and reverse (rev) primers are listed for the genome regions used in the experiment.

Primer	Sequence (5′ to 3′)	Genome Region
mt16srRNA (for)	TATCCTGACCGTGCAAAGGTA	1999–2134 bp
mt16srRNA (rev)	AGGTTATTCCAGCCTCTTCACTG	
refB2m (for)	CTTGGGCTCCTTCAGAGTGG	178,645–178,693 bp
refB2m (rev)	TGGACAAAGTCGAGCTGTCA	

**Table 3 cells-11-01763-t003:** Doubling times [h] for DP-12 FUCCI and parental CHO DP-12 for each osmolality stage for glucose-oversupplemented (Glc) and mannitol-oversupplemented (Man) feeds.

Cultivation	Doubling Times according to Osmolality Stage and Reagent [h]			
	300 mOsm/kg	370 mOsm/kg	460 mOsm/kg	530 mOsm/kg
CHO DP-12 Man	21.37	21.7	28.78	67.98
CHO DP-12 Glc	21.12	24.42	50.27	862.02
DP-12 FUCCI Man	19.3	19.67	24.26	48.54
DP-12 FUCCI Glc	21.68	24.24	35.76	195.32

## Data Availability

The data presented in this study are available on request from the corresponding author. The data are not publicly available due to the size of respective microscopic raw image data (multiple gigabytes).

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
