# Peer review of "Single-Cell Analysis of CHO Cells Reveals Clonal Heterogeneity in Hyperosmolality-Induced Stress Response"

_cells, 2022, doi:10.3390/cells11111763_

Round 1
Reviewer 1 Report
The clonal heterogeneity is very important for cell line development in industrial application of CHO cells. Cell line obtained the infinity growth during establishment from primary cell and essentially contains the heterogeneity. This paper focused on the heterogeneity of CHO cell line. Recent remarkable progress of CHO cell culture engineering in therapeutic antibody production caused the increase of osmotic pressure during fed-batch operation. This paper analyzed the heterogeneity in CHO cell under high osmotic condition. This paper is first report focusing this point and provided fundamentally important information for cell line and cell culture development.
Comments:
(1)The effect of antibody production
CHO-DP12 cells produced antibodies. If possible, it is better to show the antibody production.
(2) Osmolality measurement
The authors used a “freezing point” osmometer for the measurement of osmolality. Please mention it in the revised manuscript
(3)Single-cell analysis
The authors performed single-cell cultivation and analyzed it. I cannot understand the number of analyzed single cells. From Fig.3A and B, five cells were analyzed for each condition. Is it enough for clonal heterogeneity analysis?
(4)Table 1
Is the condition of osmolality a measurement value? The ratio of spent medium affected the results?
Author Response
(1) The effect of antibody production
CHO-DP12 cells produced antibodies. If possible, it is better to show the antibody production.
Thank you for this suggestion. We have measured antibody production changes due to high osmolality for the DP-12 cells in our previous study (Romanova 2020, https://doi.org/10.1002/bit.27747). The cell‐specific productivity qP [pg/cell × day] during the fed‐batch cultivation of CHO DP‐12 cells, based on three biological replicates per condition, shown in Figure 1 of the publication, did not show any significant changes. To test if the stage protocol used in the current article has an influence on cellular productivity, we measured punctually qP for one osmolality stage (460 mOsm/kg) for high-glucose and high-mannitol feed. The measurement was performed on day 4. Antibody production rates remained similar, as in the aforementioned publication: qP High-glucose was 9,12±0,35 pg/cell × day, qP High-mannitol 8,94±0,52 and qP control 9,52±0,65 pg/cell × day.
We have added the following text in the “Results” section
(lines 323-337):
“We punctually checked the influence of hyperosmolar treatment on cellular productivity. We have already reported that cell-specific production rate (qP) was not influenced by the exposure of DP-12 cells to the oversupplemented feed [11]. For the current protocol, we measured qP for 460 mOsm/kg osmolality stage for high-glucose, high-mannitol, and control conditions. The measurement was performed on day 4. Cell-specific antibody production rates were stable in all three conditions and were similar to those reported in the aforementioned publication: qPhigh-glucose was 9,12±0,35 pg/cell × day, qPhigh-mannitol 8,94±0,52 pg/cell × day and qPcontrol 9,52±0,65 pg/cell × day.”
(2) Osmolality measurement
The authors used a “freezing point” osmometer for the measurement of osmolality. Please mention it in the revised manuscript
The mentioning was added in the corresponding section of Matherials and Methods. The section “2.2 Stage batch cultivation in shakers” now includes following sentence (lines 160-162):
“Osmolality was measured daily by a freezing point osmometer (Gonotec GmbH, Osmomat auto, Berlin, Germany).”
(3) Single-cell analysis
The authors performed single-cell cultivation and analyzed it. I cannot understand the number of analyzed single cells. From Fig.3A and B, five cells were analyzed for each condition. Is it enough for clonal heterogeneity analysis?
We thank the reviewer for raising this matter. Since we are doing single-cell analysis, the benefit of our method lies in analyzing single cells and describing their behavior. Therefore, there is no clear advantage in increasing the number of analyzed cells drastically and determining average data for growth and morphology over the number of analyzed cells. We also validated the single cell data by calculating the mean Td (an average doubling time of the doublings completed by the tracked cells). The Td calculated based on the five cells corresponds quite well with the Td calculated for the aggregate bulk population in the shaker (e.g, untreated DP-12 cells Td single cells 21±3 h and 21.37 h for the population-level measurement, see Figure 3 and Table 3 in the result and discussion section) However, the reviewer is right to question the significance of the altered behavior of the five described cells. As we only used these data to validate in principle our heterogeneity hypothesis, we believe that the number of replicates is enough. For a more thorough investigation, which aims at detailed quantifying of these heterogeneities within the population, we agree with the reviewer's proposal that more replicates have to be analyzed. This question should be addressed in further research.
We have added following sentences in the “Conclusions” section (lines 702-706):
“The number of the tracked cells per condition (five) is sufficient for the principle validation of the heterogeneity hypothesis. A more thorough investigation aiming at detailed quantification of these heterogeneities within the population will require an analysis of significantly larger cell numbers. This question offers a direction for future research.”
(4) Table 1
Is the condition of osmolality a measurement value? The ratio of spent medium affected the results?
The conditions in the Table 1, also used throughout the manuscript to designate the stage of the treatment, are based on the mean values of the measured osmolalities for each stage aggregated for all four cultivations. For example, in the 460 mOsm/kg stage the calculated mean osmolalities based on measured values on days 1-4 for DP-12 and FUCCI cells in high-mannitol and high-glucose cultivations were 460, 456, 464, 435 mOsm/kg. We find the use of exact measured values for each cultivation to address the condition rather confusing, so we selected one representative value for each stage which is very close to the mean throughout four cultivations. We also feel that mean osmolalities are more beneficial for the reader than abstract names, such as Stage I, II etc. The detailed data on measured osmolality can be seen in the Figure A3.
I have added the following comment in the footnote of the Table 1 (lines 150-154)
“** - condition designations, e.g, 300 mOsm/kg or 370 mOsm/kg are based on approximate values of the calculated mean osmolalities through all four cultivations: CHO DP-12 and DP-12 FUCCI cells, high-mannitol and high-glucose cultivation. The exact profiles of the measured osmolalities can be seen in the Figure A3.”
Spent medium was necessary for successful single-cell cultivation, but it was added in the same proportion during the shaker cultivation as well to ensure maximal compatibility of microfluidic and batch cultivation conditions. The addition of the spent medium instead of completely fresh medium slightly decreased the measured osmolality, which otherwise lies almost exactly by 300 mOsm/kg, as is shown in the Figure A3.
Reviewer 2 Report
The authors have shown clonal heterogeneity in hyperosmolality-induced stress response in CHO cells by single-cell analysis. The work is technical sound and the authors utilized appropriate techniques of analysis. The results are supported by the data and supply useful conclusion. There are some typewriting errors and some sentences are rambling.
I only have a few comments:
-In the last part of the introduction, the Author needs to more clearly state the novelty of this paper together with future prospects of this study.
- The material presented in tables and graphics is relevant but, in my opinion figure 3, must be broken in two figures.
- Please clarify or rewrite the following sentence: “However, when the diameter increase rates are calculated (Table A1) based on tracked single cells, it becomes clear that the opposite is true: cells gain mass slower proportionally to the increase in ambient osmolality”. I do not understand, clearly the meaning
-In the result and discussion section, the author needs to pay more attention and validate their findings with recent previous results and compare if possible.
- The conclusion section must be improved to better explain the obtained results and their potentiality
Author Response
The authors have shown clonal heterogeneity in hyperosmolality-induced stress response in CHO cells by single-cell analysis. The work is technical sound and the authors utilized appropriate techniques of analysis. The results are supported by the data and supply useful conclusion. There are some typewriting errors and some sentences are rambling.
Thank you for your comments. We have checked the spelling once again in entire document.
- In the last part of the introduction, the Author needs to more clearly state the novelty of this paper together with future prospects of this study.
Thank you for your suggestion. Following text has been added into the Introduction (lines 86-100):
“Recent research of single cells revealed extensive genomic and productivity cell-to-cell heterogeneity even within populations originating from the single clone [31]. This has an extensive relevance to bioprocess performance and stability. Therefore, single-cell data on production-relevant parameters, such as growth and production rate, is in high demand [32]. The heterogeneity of stress response of suspension-grown CHO has not been previously elucidated, and there is a high demand on new insights into this aspect to investigate its impact on bioproduction processes.”
- The material presented in tables and graphics is relevant but, in my opinion figure 3, must be broken in two figures.
The Figure was broken into two: Figure 3 and Figure 4 according to your suggestion.
- Please clarify or rewrite the following sentence: “However, when the diameter increase rates are calculated (Table A1) based on tracked single cells, it becomes clear that the opposite is true: cells gain mass slower proportionally to the increase in ambient osmolality”. I do not understand, clearly the meaning
We have re-written the sentence as following and we hope, this version conveys the meaning better (lines 508-517).
However, based on single analyzed cells (Figure 3), we calculated the rates at which the cell diameter increases. This rate (Table A1) corresponds to the slope of the linear fit function (Figure 4 B) based on relative diameter data of a single randomly chosen cell in each condition. As the slope of the linear fit function decreases (Table A1, Figure 4 B), the cell diameter gain rate progressively slows down as the osmolality rises. Therefore, individual cells of the untreated population gain mass at a higher rate compared to high-mannitol and high-glucose feed exposure.
-In the result and discussion section, the author needs to pay more attention and validate their findings with recent previous results and compare if possible.
We have added comparison with other studies, where possible. Please, refer to the marked changes in the Results and Discussion section (Lines 324-328, 376-380, and 438-443).
- The conclusion section must be improved to better explain the obtained results and their potentiality
Thank you for your suggestion. We have added following into the Conclusion section (Lines 725-727 and 737-739):
Senescence induction should be possibly omitted under industrial process, therefore osmolality and nutrient availability should be kept within physiological optimum.
With these findings, our work represents the fundament for future investigations of bioprocess relevant research questions in the field of stress-induced cell-to-cell heterogeneity.